# Antiviral and clinical activity of bamlanivimab in a randomized trial of non-hospitalized adults with COVID-19

Kara W. Chew [1] ✉, Carlee Moser [2], Eric S. Daar[3], David A. Wohl [4], Jonathan Z. Li [5], Robert W. Coombs [6,7], Justin Ritz [2], Mark Giganti[2], Arzhang Cyrus Javan[8], Yijia Li[5,14], Manish C. Choudhary[5], Rinki Deo[5], Carlos Malvestutto[9], Paul Klekotka[10], Karen Price[10], Ajay Nirula[10], William Fischer[4], Veenu Bala[11,15], Ruy M. Ribeiro [12], Alan S. Perelson [12], Courtney V. Fletcher [11], Joseph J. Eron [4], Judith S. Currier [1], ACTIV-2/A5401 Study Team, Michael D. Hughes[2,16] & Davey M. Smith [13,16]

Anti-SARS-CoV-2 monoclonal antibodies are mainstay COVID-19 therapeutics. Safety, antiviral, and clinical efficacy of bamlanivimab were evaluated in the randomized controlled trial ACTIV-2/A5401. Non-hospitalized adults were randomized 1:1 within 10 days of COVID-19 symptoms to bamlanivimab or blinded-placebo in two dose-cohorts (7000 mg, $n = 94$; 700 mg, $n = 223$). No differences in bamlanivimab vs placebo were observed in the primary outcomes: proportion with undetectable nasopharyngeal SARS-CoV-2 RNA at days 3, 7, 14, 21, and 28 (risk ratio = 0.82–1.05 for 7000 mg [$p$(overall) = 0.88] and 0.81–1.21 for 700 mg [$p$(overall) = 0.49]), time to symptom improvement (median 21 vs 18.5 days [$p = 0.97$], 7000 mg; 24 vs 20.5 days [$p = 0.08$], 700 mg), or grade 3+ adverse events. However, bamlanivimab was associated with lower day 3 nasopharyngeal viral levels and faster reductions in inflammatory markers and viral decay by modeling. This study provides evidence of faster reductions in nasopharyngeal SARS-CoV-2 RNA levels but not shorter symptom durations in non-hospitalized adults with early variants of SARS-CoV-2. Trial Registration: ClinicalTrials.gov Identifier: NCT04518410.

Severe acute respiratory syndrome coronavirus 2 (SARS-CoV-2), the virus that causes Coronavirus disease 2019 (COVID-19), continues to exert an enormous global public health and economic toll, and in the U.S. case-fatality rates exceed estimates for the 1918 influenza pandemic[1]. Anti-SARS-CoV-2 monoclonal antibody (mAb)-based therapies have shown sufficient clinical efficacy to receive emergency

[1]Department of Medicine, David Geffen School of Medicine at University of California, Los Angeles, Los Angeles, CA, USA. [2]Harvard T.H. Chan School of Public Health, Boston, MA, USA. [3]Lundquist Institute at Harbor-UCLA Medical Center, Torrance, CA, USA. [4]Department of Medicine, University of North Carolina at Chapel Hill School of Medicine, Chapel Hill, NC, USA. [5]Department of Medicine, Brigham and Women's Hospital, Harvard Medical School, Boston, MA, USA. [6]Department of Laboratory Medicine and Pathology, University of Washington, Seattle, WA, USA. [7]Department of Medicine, University of Washington, Seattle, WA, USA. [8]National Institutes of Health, Bethesda, MD, USA. [9]Ohio State University Wexner Medical Center, Columbus, OH, USA. [10]Eli Lilly and Company, San Diego, CA, USA. [11]UNMC Center for Drug Discovery, University of Nebraska Medical Center, Omaha, NE, USA. [12]Theoretical Biology and Biophysics Group, Los Alamos National Laboratory, Los Alamos, NM, USA. [13]Department of Medicine, University of California, San Diego, La Jolla, CA, USA. [14]Present address: Department of Medicine, University of Pittsburgh, Pittsburgh, PA, USA. [15]Present address: Clinical Pharmacology & Pharmacometrics, Jounce Therapeutics, Cambridge, MA, USA. [16]These authors jointly supervised this work: Michael D. Hughes, Davey M. Smith. A list of members and their affiliations appears in the Supplementary Information. ✉e-mail: kchew@mednet.ucla.edu

authorization (EUA) by regulatory agencies for the treatment of COVID-19 in non-hospitalized persons[2–5].

Bamlanivimab is a neutralizing immunoglobulin G (IgG)−1 mAb directed to the receptor binding domain (RBD) of the spike (S) protein of SARS-CoV-2[6]. On November 9, 2020, based on data from the Blocking Viral Attachment and Cell Entry with SARS-CoV-2 Neutralizing Antibodies (BLAZE-1) trial (NCT04427501)[2], the Food and Drug Administration (FDA) issued an EUA for its use as a one-time 700 mg intravenous (IV) infusion for the treatment of mild to moderate COVID-19 in non-hospitalized adults with risk factors for progression to severe disease who were within 10 days of symptom onset[7]. Since then, the emergence of SARS-CoV-2 variants with decreased susceptibility in vitro to bamlanivimab[8,9] led to withdrawal of the EUA.

With the rapid development of additional anti-SARS-CoV-2 mAbs and non-antibody antivirals such as remdesivir, nirmatrelvir plus ritonavir, and molnupiravir for treatment of early COVID-19[5,10–13], understanding the antiviral activity and characterizing the clinical benefits of mAbs in this setting remains a critical need. Here, we describe the safety, virologic, and clinical outcomes of a placebo-controlled phase 2 evaluation of bamlanivimab at two doses, 7000 mg and 700 mg, in non-hospitalized adults with COVID-19. The lessons learned are relevant to clinical trial design for SARS-CoV-2 therapeutics and understanding potential mechanisms for the observed clinical benefits associated with anti-SARS-CoV-2 mAbs.

## Results

### Characteristics of participants and retention in follow-up

The analysis population included 94 participants in the bamlanivimab 7000 mg dose cohort (48 bamlanivimab, 46 placebo) enrolled between August 19 and November 15, 2020, and 223 participants in the bamlanivimab 700 mg dose cohort (111 bamlanivimab, 112 placebo) enrolled between October 12 and November 17, 2020. Across bamlanivimab and placebo arms, 3 (3.2%) of the 7000 mg group, and 6 (2.6%) of the 700 mg group prematurely discontinued the study prior to day 28 (Fig. 1, Consort Flow Diagrams).

Participant characteristics were balanced across randomized arms in both the 7000 and 700 mg groups (Table 1). Across all 317 participants included in analyses, 116 (37%) reported ≤5 days of symptoms, and 153 (48%) met the protocol definition of higher risk of progression to severe COVID-19. Hypertension, diabetes, obesity, and age were the most common high-risk criteria (Supplementary Table 1). The most

frequently reported symptoms (reported by >40% of participants) within 48 h of study entry included cough, headache, body pain or muscle pain/aches, fatigue, nasal obstruction or congestion, nasal discharge, and loss of taste or smell (Supplementary Table 2); most symptoms were reported as mild or moderate.

### Safety

Treatment-emergent adverse events (TEAEs) through study day 28 are summarized in Table 2 and TEAEs through week 24 in Supplementary Table 3. Grade 2 or higher and grade 3 or higher TEAEs were generally more frequently reported in bamlanivimab 700 and 7000 mg recipients than in placebo recipients, but the proportion with grade 3 or higher TEAEs (the primary safety outcome) did not differ significantly between bamlanivimab vs placebo arms for either dose and the vast majority of TEAEs were not felt to be related to study intervention. Adverse events of special interest (AESIs) were infrequent and led to premature treatment discontinuation in only one participant assigned bamlanivimab 7000 mg, who did not complete the infusion due to a grade 3 infusion-related reaction (IRR). Serious adverse events (SAEs) through day 28 occurred in 2 (4.2%) and 4 (8.7%) of bamlanivimab 7000 mg and placebo recipients, respectively, and in 4 (3.6%) and 3 (2.7%) of bamlanivimab 700 mg and placebo recipients, respectively (Table 2). Detailed summaries of grade 2 and higher and grade 3 and higher TEAEs through day 28 by dose cohort and treatment arm are provided in Supplementary Tables 4 and 5.

### Virological and related outcomes

SARS-CoV-2 variant determination was successful in 77 of 78 participants in the 7000 mg dose cohort (42 in bamlanivimab and 35 in placebo arms) and in 207 of 208 participants in the 700 mg dose cohort (101 in bamlanivimab and 106 in placebo arms), with 1 sequencing failure in each 7000 mg and 700 mg placebo arm. Three participants in the 700 mg dose cohort were infected with the Epsilon variant (1 in bamlanivimab and 2 in placebo arm). No other variants of concern or variants of interest were identified (Supplementary Table 6).

Baseline nasopharyngeal (NP) SARS-CoV-2 RNA levels were similar at study entry between bamlanivimab vs placebo arms in each dose cohort (Fig. 2 and Table 3). The proportion of participants with undetectable NP SARS-CoV-2 RNA (primary virologic outcome) increased over time and did not differ between bamlanivimab or placebo arms for either dose cohort (Fig. 2 and Table 3). At day 3, the

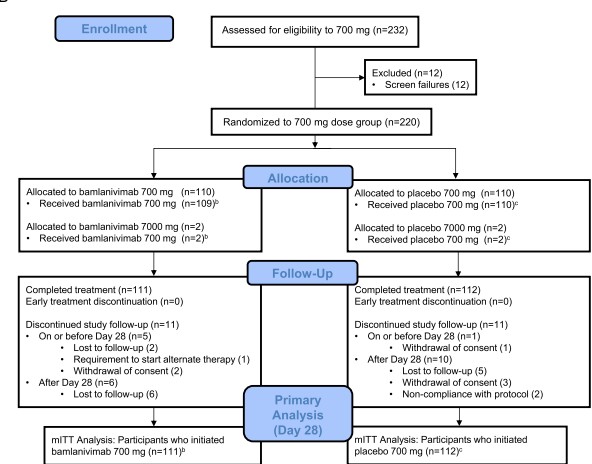

**Fig. 1 | CONSORT flow diagram for each bamlanivimab dose cohort. A** 7000 mg dose cohort and **B** 700 mg dose cohort. [a]Of the $n$ = 48 participants that received bamlanivimab 7000 mg, one participant was assigned to bamlanivimab 700 mg, but received bamlanivimab 7000 mg. [b]Of the $n$ = 111 participants that received bamlanivimab 700 mg, 2 participants were assigned bamlanivimab 7000 mg, but

received bamlanivimab 700 mg. [c]Of the $n$ = 112 participants that received placebo for bamlanivimab 700 mg, 2 participants were assigned placebo for bamlanivimab 7000 mg (the placebo was the same for both bamlanivimab 7000 mg and 700 mg, normal saline).

**Table 1 | Baseline participant characteristics by dose cohort and treatment arm**

| Characteristic | 7000 mg dose cohort | | | 700 mg dose cohort | | |
|---|---|---|---|---|---|---|
| | Bamlanivimab (N = 48) | Placebo (N = 46) | Total (N = 94) | Bamlanivimab (N = 111) | Placebo (N = 112) | Total (N = 223) |
| Age, years, median (IQR) | 45.5 (33.5, 57.5) | 42.0 (28.0, 54.0) | 44.5 (30.0, 56.0) | 46.0 (35.0, 54.0) | 48.5 (36.0, 55.0) | 47.0 (35.0, 55.0) |
| **Sex, n (%)** | | | | | | |
| Female | 26 (54.2) | 23 (50.0) | 49 (52.1) | 57 (51.4) | 56 (50.0) | 113 (50.7) |
| Male | 22 (45.8) | 23 (50.0) | 45 (47.9) | 54 (48.6) | 56 (50.0) | 110 (49.3) |
| **Race, n (%)** | | | | | | |
| White | 41 (85.4) | 37 (80.4) | 78 (83.0) | 92 (82.9) | 92 (82.9) | 184 (82.9) |
| Black | 3 (6.3) | 2 (4.3) | 5 (5.3) | 13 (11.7) | 10 (9.0) | 23 (10.4) |
| Asian | 0 (0) | 5 (10.9) | 5 (5.3) | 2 (1.8) | 4 (3.6) | 6 (2.7) |
| Other[a] | 4 (8.3) | 2 (4.3) | 6 (6.4) | 4 (3.6) | 5 (4.5) | 9 (4.0) |
| Missing | 0 | 0 | 0 | 0 | 1 | 1 |
| **Ethnicity, n (%)** | | | | | | |
| Hispanic/Latino | 15 (31.3) | 17 (37.8) | 32 (34.4) | 18 (16.2) | 31 (28.2) | 49 (22.2) |
| Not Hispanic/Latino | 33 (68.8) | 28 (62.2) | 61 (65.6) | 93 (83.8) | 79 (71.8) | 172 (77.8) |
| Missing | 0 | 1 | 1 | 0 | 2 | 2 |
| Days from symptom onset at study entry (IQR) | 6.0 (4.0, 8.0) | 5.5 (4.0, 7.0) | 6.0 (4.0, 7.0) | 6.0 (4.0, 8.0) | 6.0 (4.0, 7.0) | 6.0 (4.0, 8.0) |
| ≤5 days, n (%) | 17 (35.4) | 17 (37.0) | 34 (36.2) | 41 (36.0) | 41 (36.6) | 82 (36.8) |
| >5 days, n (%) | 31 (64.6) | 29 (63.0) | 60 (63.8) | 70 (63.1) | 71 (63.4) | 141 (63.4) |
| **Risk of COVID-19 progression, n (%)** | | | | | | |
| Higher risk | 20 (41.7) | 19 (41.3) | 39 (41.5) | 58 (52.3) | 56 (50.0) | 114 (51.1) |
| Lower risk | 28 (58.3) | 27 (58.7) | 55 (58.5) | 53 (47.7) | 56 (50.0) | 109 (48.9) |
| BMI (kg/m$^2$), median (IQR) | 28.2 (24.9, 31.8) | 28.8 (25.0, 31.3) | 28.5 (25.0, 31.8) | 28.4 (25.1, 33.9) | 27.1 (23.8, 32.1) | 27.8 (24.5, 32.9) |
| Missing | 6 | 7 | 13 | 16 | 15 | 31 |

[a]Other includes Asian, American Indian or Alaskan, multiple races, and other race.
*IQR* interquartile range, *BMI* body mass index.

median NP SARS-CoV-2 RNA level was significantly lower among bamlanivimab 700 mg recipients compared to placebo (2.9 vs 3.9 log$_{10}$ copies/mL, $p = 0.002$), and a similar trend was observed for bamlanivimab 7000 mg compared to placebo (2.2 vs 3.4 log$_{10}$ copies/mL, $p = 0.07$). No differences in SARS-CoV-2 RNA levels between bamlanivimab and placebo arms were observed at any of the later visits. Additionally, the area under the curve (AUC) for SARS-CoV-2 RNA from day 0 through day 28 was smaller for both bamlanivimab 700 mg and 7000 mg compared to placebo, but neither difference met statistical significance (Table 3).

Examining NP SARS-CoV-2 RNA levels stratifying participants by time from symptom onset at study entry (≤5 vs >5 days), participants with ≤5 days of symptoms had higher SARS-CoV-2 RNA levels at entry and larger differences in SARS-CoV-2 RNA levels at day 3 favoring bamlanivimab (difference in medians of −1.4 log$_{10}$ copies/mL for bamlanivimab vs placebo) than those who entered the study >5 days from symptom onset (difference in medians of −0.9 log$_{10}$ copies/mL) for the 700 mg dose cohort, with similar trends observed for the smaller 7000 mg dose cohort (Supplementary Fig. 1 and Supplementary Table 7).

The SARS-CoV-2 RNA level (viral load) decay from NP and anterior nasal (AN) swab data was fitted for those participants for whom there was enough data (Supplementary Table 8A and Supplementary Fig. 2 and Supplementary Fig. 3). Decay rates were similar for each dose cohort (Supplementary Table 8B); thus, the 700 and 7000 mg dose cohorts were combined and fitted together, with separate analyses for NP and AN data. Population parameter estimates for the viral load decay in NP and AN swabs are provided in Supplementary Table 8C. The best model for AN data had a single exponential decay, and for the NP data, a biexponential decay. The first phase of viral decay was fast (AN: t$_{1/2}$ = 7.8 and 6.5 hours and NP: t$_{1/2}$ = 10.3 and 7.2 h for placebo and bamlanivimab-treated participants, respectively), while the second

phase was slightly slower with t$_{1/2}$ = 15.1 h (in NP), with no difference in the second phase decay rate between placebo and bamlanivimab-treated participants. In both AN and NP models, the first phase of decay was significantly faster ($p = 0.0049$ and $p = 0.0002$, respectively) for bamlanivimab treatment compared to placebo. No difference was observed in decay rates comparing participants who were treated within 5 days vs >5 days from symptom onset on either NP or AN swabs (Supplementary Fig. 4).

Pooling both dose cohorts, baseline NP and AN viral load were highly correlated ($r = 0.85$, $p < 0.001$) (Supplementary Fig. 5A), with AN viral load lower than NP viral load (median [interquartile range, IQR] 4.5 [2.3, 6.3] and 5.5 [3.6, 6.8] log$_{10}$ copies/mL for AN and NP swabs, respectively). Baseline NP and AN viral loads did not differ by risk category for COVID-19 progression (median [IQR] NP viral load 5.4 [3.5, 6.8] in higher risk vs 5.5 [3.7, 6.7] log$_{10}$ copies/mL in lower risk, $p = 0.8$) (Supplementary Figure 5B), but were higher among plasma SARS-CoV-2 viremic vs aviremic participants (median NP viral load [IQR] 6.4 [5.3, 7.6] vs 5.3 [3.3, 6.6], $p < 0.0001$) (Supplementary Figure 5C). Twenty percent of participants had detectable plasma SARS-CoV-2 RNA at baseline, without difference in proportion viremic by risk category for COVID-19 progression (Supplementary Figure 5D). Total symptom score was higher among viremic vs aviremic participants at baseline (Supplementary Figure 5E); symptom score did not correlate with NP or AN viral load (Figure 5F).

**Symptoms and other clinical outcomes**

Overall, time to symptom improvement (primary symptom outcome) was long and did not differ significantly between bamlanivimab vs placebo arms for either dose cohort (median of 21 and 18.5 days for bamlanivimab 7000 mg vs placebo, $p = 0.97$ and 24 vs 20.5 days for bamlanivimab 700 mg vs placebo, $p = 0.08$) (Table 4). AUC of the total symptom score reported daily days 0-28 in the study diary also did not

**Table 2 | Adverse events (AEs) through day 28**

| Event | 7000 mg dose cohort | | | | 700 mg dose cohort | | | |
|---|---|---|---|---|---|---|---|---|
| | Bamlanivimab (n = 48) | Placebo (n = 46) | Risk Ratio (bamlanivimab vs placebo) (95% CI), p-value[a] | Risk difference (bamlanivimab vs placebo) (95% CI) | Bamlanivimab (n = 111) | Placebo (n = 112) | Risk Ratio (bamlanivimab vs placebo) (95% CI), p-value[a] | Risk difference (bamlanivimab vs placebo) (95% CI) |
| Grade 3 or higher TEAEs through day 28 (primary safety outcome), number of participants (%) | 6 (12.5) | 6 (13.0) | 0.96 (0.33, 2.76), p = 0.94 | −0.5 (−14.0, 13.0) % | 12 (10.8) | 7 (6.3) | 1.73 (0.71, 4.23), p = 0.23 | 4.6 (−2.8, 11.9) % |
| Grade 2 or higher TEAEs through day 28, number of participants (%) | 20 (41.7) | 16 (34.8) | 1.20 (0.71, 2.01), p = 0.49 | 6.9 (−12.7, 26.5) % | 49 (44.1) | 31 (27.7) | 1.59 (1.11, 2.30), p = 0.01 | 16.5 (4.1, 28.9) % |
| AEs leading to premature treatment discontinuation, number of participants (%) | 1 (2.1) | 0 | – | – | 0 | 0 | – | – |
| AESIs through day 28, number of participants (%) | 1 (2.1) | 2 (4.3) | – | – | 1 (0.9) | 3 (2.7) | – | – |
| Infusion-related reaction | 1 (2.1) | 1 (2.2) | – | – | 1 (0.9) | 1 (0.9) | – | – |
| Hypersensitivity reaction | 0 | 1 (2.2) | – | – | 0 | 2 (1.8) | – | – |
| Serious adverse events (SAEs) through day 28, number of participants (%) | 2 (4.2) | 4 (8.7) | – | – | 4 (3.6) | 3 (2.7) | – | – |

*TEAE treatment emergent adverse event, AESI adverse event of special interest, [a]two-sided Wald chi-square test.*

differ significantly between bamlanivimab vs placebo arms for either dose cohort (Table 4).

Through study day 28, there were 6 hospitalizations in the 7000 mg group, 2 (4.2%) on bamlanivimab and 4 (8.7%) on placebo, and 8 hospitalizations in the 700 mg group, 4 (3.6%) on bamlanivimab and 4 (3.6%) on placebo. No deaths were observed through week 24. Hospitalizations and deaths through week 24 are summarized in Supplementary Table 9.

C-reactive protein (CRP), ferritin, and fibrinogen levels declined more rapidly (greater fold change from baseline) in bamlanivimab 700 mg compared to placebo recipients at days 7 and 14 (as well as week 24 for CRP) (Supplementary Fig. 6). Greater fold-change reductions in prothrombin time (PT) were also observed at Days 14, 21, and Week 12 with bamlanivimab 700 mg. Similar trends were observed at some time points for bamlanivimab 7000 mg vs placebo (Supplementary Fig. 7). No differences between bamlanivimab vs placebo arms were observed for fold change from baseline for lactate dehydrogenase (LDH) or activated partial thromboplastin time (aPTT) through day 28 (Supplementary Figs. 6 and 7).

## Pharmacokinetics

PK data were obtained on a total of 108 participants (71 who received 700 mg and 37 who received 7000 mg), as summarized in Supplementary Table 10. Mean Cmax values for the 700 mg and 7000 mg doses were 206 and 1876 µg/mL, respectively, and mean day 28 concentrations were 29 and 236 µg/mL, respectively. There was evidence for approximate dose proportionality, with geometric mean ratios for Cmax and $AUC_{0-\infty}$ of 8.3 and 8.5, respectively; the geometric mean ratio for total body clearance (CL) was 1.2. Interpatient variability was modest with coefficients of variation (CV) on CL of 40.5% at the 700 mg dose and 88.9% at the 7000 mg dose. Of bamlanivimab 700 mg recipients, 70/71 (98.6%) had day 28 concentrations above the estimated 90% inhibitory concentration (IC90) of bamlanivimab for SARS-CoV-neutralization of 4.2 µg/mL[14]; one participant had bamlanivimab concentrations below the limit of quantitation at day 28.

## Discussion

We present results of a placebo-controlled phase 2 evaluation of the safety and efficacy of single dose bamlanivimab 700 mg and partially enrolled phase 2 evaluation of single dose bamlanivimab 7000 mg given by IV infusion for the treatment of non-hospitalized adults with COVID-19. Consistent across both dose cohorts, in which participants received study intervention a median of 6 days from symptom onset, bamlanivimab was safe and reduced NP SARS-CoV-2 RNA levels and circulating measures of inflammation more rapidly than placebo. We also demonstrate that antiviral activity of mAbs in the nasal compartment may be observed with treatment out to 10 days from symptom onset and identify key limitations of NP SARS-CoV-2 RNA measures for assessing the antiviral activity of COVID-19 therapeutics.

We found that, among participants who were enrolled within 5 days of symptom onset, NP SARS-CoV-2 RNA levels were >1.5 $\log_{10}$ copies/mL higher at study entry than among those enrolled >5 days from symptom onset and differences in early post-treatment NP SARS-CoV-2 RNA levels favoring bamlanivimab were more pronounced amongst those treated within 5 days of symptom onset. Modeled viral decay rates were not different between those enrolling earlier versus later, with both groups showing more rapid decline with bamlanivimab than placebo. These findings highlight the limitations of nasal compartment SARS-CoV-2 RNA measures for the evaluation of SARS-CoV-2 treatment interventions, particularly when examining absolute levels or changes in levels from baseline, as has been done in multiple randomized clinical trials of mAbs and oral antivirals for early COVID-19 treatment. In these trials, both significant reductions (with bamlanivimab, bamlanivimab/etesevimab, casirivimab/imdevimab, and

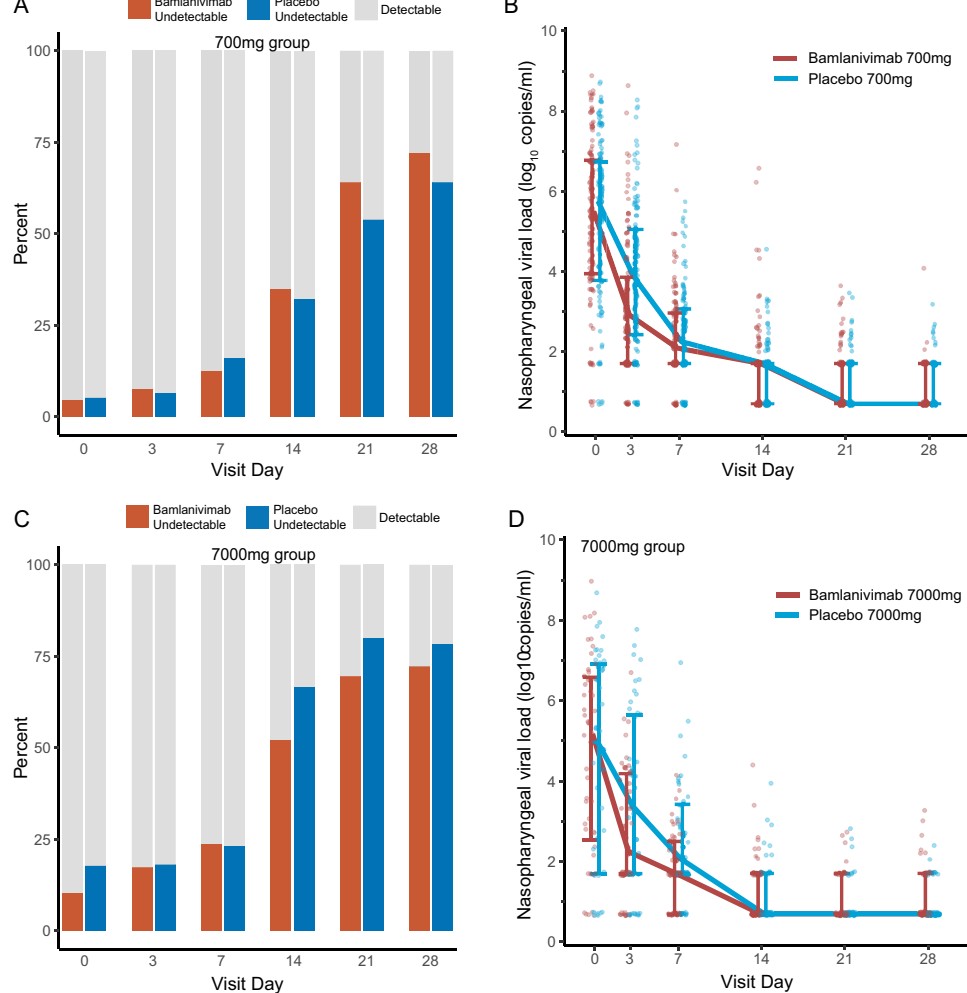

**Fig. 2 | Nasopharyngeal (NP) SARS-CoV-2 RNA levels (viral loads) by dose cohort, treatment arm, and visit.** NP viral loads declined in all participants, without a difference in proportion undetectable at any time points (primary virologic outcome) for either the 700 mg bamlanivimab dose (**A**) or the 7000 mg dose (**C**), with risk ratio (RR) [95% confidence interval, CI] for non-detection of SARS-CoV-2 RNA for bamlanivimab vs placebo in the 700 mg dose cohort (bamlanivimab $n = 111$, placebo = 112) of 1.21 (0.48, 3.06) at day 3 and 0.81 (0.43, 1.53) at day 7, or the 7000 mg dose cohort (bamlanivimab $n = 48$, placebo = 46), 0.97 (0.44, 2.16) at day 3 and 1.05 (0.53, 2.08) at day 7 (see Table 3 for risk ratios at later time points). Risk ratios and 95% CI were calculated by Poisson regression model for repeated measures with robust variance and log-link fit with generalized estimating equations

with an independence working correlation structure. Median NP viral loads were lower at day 3 for bamlanivimab 700 mg vs placebo (2.9 vs 3.9 $\log_{10}$ copies/mL, $p = 0.002$) (**B**), with similar findings seen, though not statistically significant, for the smaller 7000 mg dose cohort (2.2 vs 3.4 $\log_{10}$ copies/mL, $p = 0.07$) (**D**). See Table 3 for quantitative SARS-CoV-2 RNA levels at later time points. P-values for comparison of median NP viral loads were generated by two-sided Wilcoxon tests. No adjustment was made for multiple comparisons. The lower limit of detection was 1.4 $\log_{10}$ copies/mL. Presented in **B** and **D** are median values with error bars for interquartile ranges, and individual participant values as dots. The bamlanivimab arm is represented by the color orange and placebo by the color blue. Source Data for summary measures are provided as a Source Data file.

nirmatrelvir/ritonavir)[2–4,12,15] and absence of significant reductions (with remdesivir and sotrovimab)[5,11,16] in NP SARS-CoV-2 RNA have been found despite evidence of a large beneficial effect of treatment on hospitalizations and deaths. Our data suggest that NP RNA shedding analyzed as such to measure antiviral activity is likely to be most informative early after symptom onset, and that modeling of viral decay rates may be a more consistent measure of antiviral activity; the association of decay rates with hospitalization and death outcomes should be explored.

In addition to reductions in NP SARS-CoV-2 RNA levels, we found biological evidence of activity of bamlanivimab against COVID-19 progression, with greater reductions in inflammatory biomarker levels (CRP, ferritin, and fibrinogen) with bamlanivimab compared to placebo. Greater reductions in CRP in the first four weeks following effective treatment were recently reported in a randomized placebo-controlled trial of molnupiravir (the MOVe-OUT trial) for COVID-19 treatment in non-hospitalized persons[17]. Our findings are consistent with these and extend them to other markers of inflammation. Our

findings also suggest potential benefits of antiviral therapies beyond the first few weeks after treatment, as we observed benefits in CRP as late as 24 weeks after bamlanivimab treatment (measures beyond day 29 were not reported for the MOVe-OUT trial). Considering the different mechanisms of action and half-life of bamlanivimab and molnupiravir, these data suggest the initial improvement in inflammatory markers is due to direct virus neutralization and inhibition of further viral replication. However, one could consider if there might be further benefit specific to mAbs that is not related to direct antiviral activity, such as through non-neutralizing Fc-mediated effector functions[18] and clearance of infected cells, that could theoretically induce greater and more prolonged benefits in SARS-CoV-2-associated inflammation and complications; a head-to-head study of small molecule antivirals with antibody-based therapies would be needed to explore this. As more potent oral antivirals for COVID-19 are becoming increasingly available, future studies are needed to interrogate the potential benefits of small molecule vs antibody-based antivirals for populations at high risk

**Table 3 | Primary and secondary virological outcomes (nasopharyngeal SARS-CoV-2 RNA) by bamlanivimab dose cohort and treatment arm**

| | 7000 mg dose cohort | | | | 700 mg dose cohort | | | |
|---|---|---|---|---|---|---|---|---|
| Primary virological outcome: Proportion NP SARS-CoV-2 RNA not detected | Bamlanivimab (N=48) NP SARS-CoV-2 RNA not detected, n (%) | Placebo (N=46) NP SARS-CoV-2 RNA not detected, n (%) | Risk ratio, bamlanivimab vs placebo (95% CI)[a] | Risk difference, bamlanivimab vs placebo (95% CI) | Bamlanivimab (N=111) NP SARS-CoV-2 RNA not detected, n (%) | Placebo (N=112) NP SARS-CoV-2 RNA not detected, n (%) | Risk ratio, bamlanivimab vs placebo (95% CI)[a] | Risk difference, bamlanivimab vs placebo (95% CI) |
| Day 0 | 5 (10.4) | 8 (18.2) | – | – | 5 (4.6) | 6 (5.4) | – | – |
| Missing | 0 | 2 | | | 2 | 0 | | |
| Day 3 | 8 (17.4) | 8 (18.2) | 0.97 (0.44, 2.16) | –0.01 (–0.17, 0.15) | 8 (7.6) | 7 (6.5) | 1.21 (0.48, 3.06) | 0.01 (–0.06, 0.08) |
| Missing | 2 | 2 | | | 6 | 5 | | |
| Day 7 | 11 (23.9) | 10 (23.3) | 1.05 (0.53, 2.08) | 0.01 (–0.17, 0.18) | 13 (12.5) | 17 (16.0) | 0.81 (0.43, 1.53) | –0.04 (–0.13, 0.06) |
| Missing | 2 | 3 | | | 7 | 6 | | |
| Day 14 | 24 (52.2) | 28 (66.7) | 0.82 (0.60, 1.12) | –0.14 (–0.35, 0.06) | 35 (35.0) | 33 (32.4) | 1.09 (0.75, 1.57) | 0.03 (–0.10, 0.16) |
| Missing | 2 | 4 | | | 11 | 10 | | |
| Day 21 | 32 (69.6) | 32 (80.0) | 0.91 (0.71, 1.15) | –0.10 (–0.29, 0.08) | 66 (64.1) | 56 (53.8) | 1.20 (0.95, 1.51) | 0.10 (–0.03, 0.24) |
| Missing | 2 | 6 | | | 8 | 8 | | |
| Day 28 | 34 (72.3) | 33 (78.6) | 0.98 (0.76, 1.25) | –0.06 (–0.24, 0.12) | 70 (72.2) | 66 (64.1) | 1.13 (0.93, 1.39) | 0.08 (–0.05, 0.21) |
| Missing | 1 | 4 | | | 14 | 9 | | |
| Overall p-value[b] | | | 0.88 | | | | 0.49 | |
| Secondary outcome: Quantitative NP SARS-CoV-2 RNA levels | NP SARS-CoV-2 RNA level, Median (Q1, Q3), log10 copies/mL | NP SARS-CoV-2 RNA level, Median (Q1, Q3), log10 copies/mL | p-value[c] | | NP SARS-CoV-2 RNA level, Median (Q1, Q3), log10 copies/mL | NP SARS-CoV-2 RNA level, Median (Q1, Q3), log10 copies/mL | p-value[c] | |
| Day 0 | 5.2 (2.5, 6.6) (n=48) | 5.0 (1.7, 6.9) (n=45) | – | | 5.5 (4.0, 6.8) (n=109) | 5.7 (3.8, 6.8) (n=112) | – | |
| Day 3 | 2.2 (1.7, 4.2) (n=46) | 3.4 (1.7, 5.7) (n=44) | p=0.07 | | 2.9 (1.7, 3.9) (n=105) | 3.9 (2.4, 5.1) (n=107) | p=0.002 | |
| Day 7 | 1.7 (0.7, 2.5) (n=46) | 2.1 (1.7, 3.4) (n=43) | p=0.14 | | 2.1 (1.7, 3.0) (n=104) | 2.2 (1.7, 3.1) (n=106) | p=0.89 | |
| Day 14 | 0.7 (0.7, 1.7) (n=46) | 0.7 (0.7, 1.7) (n=42) | p=0.27 | | 1.7 (0.7, 1.7) (n=100) | 1.7 (0.7, 1.7) (n=102) | p=0.90 | |
| Day 21 | 0.7 (0.7, 1.7) (n=46) | 0.7 (0.7, 0.7) (n=40) | p=0.31 | | 0.7 (0.7, 1.7) (n=103) | 0.7 (0.7, 1.7) (n=104) | p=0.29 | |
| Day 28 | 0.7 (0.7, 1.7) (n=47) | 0.7 (0.7, 0.7) (n=42) | p=0.40 | | 0.7 (0.7, 1.7) (n=97) | 0.7 (0.7, 1.7) (n=103) | p=0.17 | |
| Secondary outcome: AUC of NP SARS-CoV-2 RNA from day 0 through day 28 | AUC of NP SARS-CoV-2 RNA, Median (Q1, Q3) | AUC of NP SARS-CoV-2 RNA, Median (Q1, Q3) | p-value[c] | | AUC of NP SARS-CoV-2 RNA, Median (Q1, Q3) | AUC of NP SARS-CoV-2 RNA, Median (Q1, Q3) | p-value[c] | |
| AUC (days x days x log10 copies/mL) | 6.4 (1.9, 17.1) (n=48) | 13.5 (1.4, 28.8) (n=45) | p=0.14 | | 12.6 (7.6, 22.4) (n=109) | 15.8 (7.6, 25.8) (n=112) | p=0.19 | |

CI confidence interval, NP nasopharyngeal, RNA ribonucleic acid, AUC area under the curve of log10 RNA and above the assay lower limit of quantification (2 log10 copies/mL). [a]Log-binomial model failed to converge thus Poisson regression model for repeated measures with robust variance and log-link fit with generalized estimating equations with an independence working correlation structure used to generate RRs and 95% CIs, [b]two-sided Wald chi-square test, [c]two-sided Wilcoxon–Mann–Whitney test.

**Table 4 | Symptom outcomes by bamlanivimab dose cohort and treatment arm**

| | 7000 mg dose cohort | | | 700 mg dose cohort | | |
|---|---|---|---|---|---|---|
| | Bamlanivimab (N = 48) | Placebo (N = 46) | p-value | Bamlanivimab (N = 111) | Placebo (N = 112) | p-value |
| Time to symptom improvement from study entry (primary symptom outcome), median (IQR), days | 21.0 (7.0, 28.0) | 18.5 (7.0, 28.0) | 0.97[a] | 24.0 (14.0, 28.0) | 20.5 (9.0, 28.0) | 0.08[a] |
| Proportion of participants with at least 1 symptom reported as more severe than at study entry in study diary, n (%) | 42 (87.5) | 40 (87.0) | 0.94[b] | 102 (91.9) | 105 (93.8) | 0.59[b] |
| Symptom severity ranking (AUC of total symptom score days 0–28), median (IQR) | 1.38 (0.93, 3.09) | 1.88 (1.09, 3.05) | 0.14[a] | 2.34 (1.30, 3.93) | 2.13 (1.06, 4.08) | 0.65[a] |

IQR interquartile range, [a] two-sided Wilcoxon test, [b] two-sided Wald chi-square test.

of severe COVID-19 (immunocompromised persons with limited endogenous immune responses) and for potential inflammation-mediated complications in all persons at risk for COVID-19, such as post-acute sequelae of SARS-CoV-2 infection (PASC). The reductions in inflammatory biomarkers with bamlanivimab, including as late as 24 weeks after treatment, suggest some promise for early antiviral therapy to mitigate or prevent the development of PASC, or some PASC manifestations.

While no impact of bamlanivimab therapy on symptom duration was found in our study, we note that symptom-based outcome measures for assessing treatment response have not yet been validated and persistence or brief recurrence of mild symptoms may have made our primary symptom outcome definition overly sensitive to symptoms that may not have been clinically meaningful. Across outpatient COVID-19 therapeutic studies, definitions of symptom improvement or resolution, symptom diaries, severity scoring scales, analytical approaches, and included symptoms have differed, yielding differing effect sizes ranging from modest (1 day reduction in time to symptom resolution) in the BLAZE-1 study[2,15] to 4 days with casirivimab/imdevimab[10]. One obvious impact of the different definitions of symptom resolution is on the duration of symptoms from study entry – shorter durations were reported in the BLAZE-1 and REGEN-COV studies than in our study (median durations of >20 days). Based on clinical observations that COVID-19 symptoms may wax and wane day-to-day, we sought to minimize the possibility of recurrent or relapsing symptoms in defining our population with improved symptoms, resulting in long durations of symptoms. An additional challenge is distinguishing between symptoms that are specific to COVID-19 or due to comorbidities, given many viral illness symptoms are non-specific. The potential for mAbs to modify symptom duration and accelerate symptom resolution may also depend on timing of mAb therapy during the COVID-19 disease course – where later administration (as late as 10 days after symptom onset, as in our study and past mAb EUA guidance) may have less impact, although this is unknown. Validation of COVID-19 symptom diary content and outcome measures are ongoing, by our group and others.

Our study yielded information on virus shedding in nasal and plasma compartments in early COVID-19 – we found NP and AN SARS-CoV-2 RNA levels are not associated with risk factors for disease progression or correlated with self-reported symptom scores but are higher among persons with SARS-CoV-2 viremia compared to those without detectable viremia. Symptom scores also tended to be higher among viremic compared to aviremic participants, suggesting plasma viral measures may be more specific measures of symptom burden or disease severity. Given the small number of hospitalizations, we could not determine if nasal shedding, viremia, or changes with treatment were associated with COVID-19 severity, as has been observed in hospitalized persons[19,20].

In this study, we also expand on the reported PK of bamlanivimab and found the characteristics in 108 non-hospitalized persons were comparable with those reported in the first-in-human study in hospitalized patients for both the 700 and 7000 mg doses[21]. Absolute dose proportional PK were not observed, explained by the faster total body clearance with 7000 mg than 700 mg. The 700 mg dose (the dose that was granted an EUA) achieved sustained serum concentrations above the predicted $IC_{90}$ for SARS-CoV-2 neutralization for authentic and early SARS-CoV-2 variants in nearly all participants with available PK data, supporting the selection of this dose for clinical use. However, bamlanivimab concentrations in target tissues such as the respiratory tract have not been measured in humans and whether this same pharmacodynamic relationship exists in tissues is not known.

As noted throughout this report, this study has several limitations including small sample size for the 7000 mg dose cohort and for higher risk participants and low rate of hospitalization/death events, which precludes analysis of the clinical significance of virologic and

inflammatory marker changes with respect to progression to severe COVID-19. The study also evaluated an agent that no longer has sufficient antiviral activity against circulating variants. However, while bamlanivimab and, increasingly, other anti-SARS-CoV-2 mAbs that have received FDA EUA do not currently have clinical utility based on the lack of in vitro neutralizing activity against variants that have emerged over the course of the COVID-19 pandemic, these phase 2 trial results affirm the overall safety and antiviral activity of bamlanivimab and offer insights into the potential benefits of mAbs and COVID-19 clinical trial design. Our study also suggests a benefit of mAbs on systemic inflammation, demonstrates that timing of COVID-19 treatment can markedly impact nasal compartment SARS-CoV-2 RNA outcomes, and highlights key areas for future work in SARS-CoV-2 therapeutics for early COVID-19, including exploration of alternate virologic measures such as viral decay rates as a predictor or surrogate for hospitalizations and deaths in COVID-19 treatment trials, and further exploration of mechanisms by which mAbs may uniquely, distinct from direct antivirals and based on individual mAb design, serve an important role in improving both early and late COVID-19 outcomes.

## Methods

### Trial design and oversight
The ACTIV-2/A5401 study is an ongoing multicenter phase 2/3 adaptive platform randomized controlled trial for the evaluation of therapeutics for early COVID-19 in non-hospitalized adults (see Supplementary Methods for the ACTIV-2/A5401 protocol). Phase 2 results for bamlanivimab compared to placebo are reported here. All participants for this phase 2 analysis were enrolled in the U.S., across 38 sites (listed in Supplementary Notes). The protocol was approved by a central institutional review board (IRB), Advarra (Pro00045266), with additional local IRB review and approval as required by participating sites. In addition to central IRB approval, the following institutions' IRB also reviewed and approved the protocol: Case Western Reserve University, Weill Cornell Medicine, Johns Hopkins University, Rush University Medical Center, and Vanderbilt University. All other participating sites obtained central IRB approval only or the institution agreed to rely on Advarra review. All participants provided written informed consent.

An independent Data Safety and Monitoring Board (DSMB) reviewed interim safety during phase 2 for both doses. There were no predefined interim stopping guidelines for phase 2.

### Participants
Adults 18 years of age or older with documented SARS-CoV-2 infection by an FDA-authorized antigen or nucleic acid test from a sample collected within 7 days prior to anticipated study entry, no more than 10 days of COVID-19 symptoms at time of anticipated study entry, ongoing symptoms (not including loss of taste or smell) within 48 h prior to study entry, resting peripheral oxygen saturation levels ≥92%, and without the need for hospitalization were eligible. Complete eligibility criteria are provided in the protocol in Supplementary Methods.

### Randomization
Participants were randomly assigned by a web-based interactive response system in a 1:1 ratio to receive either blinded bamlanivimab or placebo using random block size of four. Randomization was stratified by time from symptom onset (≤ or >5 days) and risk of progression to severe COVID-19 (higher vs lower). Higher risk was defined in the protocol as meeting any of the following: age ≥55 years or having a comorbidity (chronic lung disease or moderate to severe asthma, body mass index >35 kg/m$^2$, hypertension, cardiovascular disease, diabetes, or chronic kidney or liver disease). Site staff and investigators, with the exception of unblinded pharmacists, were blind to randomized treatment.

### Study intervention
An initial dose of 7000 mg of bamlanivimab was chosen for study based on PK and preliminary safety data. After phase 2 data on bamlanivimab 700 mg, 2800 mg, and 7000 mg from an outside trial did not conclusively demonstrate a dose-response effect of higher doses on declines in NP SARS-CoV-2 RNA levels, the protocol was amended to evaluate the 700 mg dose prior to completing planned enrollment[2]. Bamlanivimab or placebo (normal saline) was prepared by unblinded site pharmacists and administered by blinded site staff as a single IV infusion over approximately 60 minutes. The IV bags and primed infusion sets were encased in opaque covers to conceal intervention.

### Primary and secondary outcomes
The phase 2 study was designed to evaluate the safety of bamlanivimab and determine the efficacy of bamlanivimab to reduce the duration of COVID-19 symptoms and SARS-CoV-2 RNA shedding from NP swabs. NP swabs were collected on days 0 (day of study intervention, pre-intervention), 3, 7, 14, 21, and 28. Participants completed a study diary each day from day 0 to day 28, which included self-assessment of 13 targeted COVID-19 symptoms, graded by the participant as absent, mild, moderate, or severe (see Supplementary Methods for symptom diary). A numerical total symptom score was calculated for each day by summing scores for each symptom, with absent scored as 0, mild as 1, moderate as 2, and severe as 3; therefore, the range of total symptom scores was 0 to 39. Clinical assessments for adverse events were conducted at days 0, 2, 3, 7, 10, 14, 21, 28, week 12, and week 24. Safety laboratories were performed at days 0, 3, 14, and 28, and included complete blood cell count with automated differential and platelet count, liver, and kidney function tests. All safety laboratories were performed at a central laboratory, PPD® Laboratory Services Global Central Labs. Adverse events including laboratory values were assessed and graded using the Division of AIDS Table for Grading the Severity of Adult and Pediatric Adverse Events, corrected Version 2.1, July 2017 (https://rsc.niaid.nih.gov/clinical-research-sites/daids-adverse-event-grading-tables).

Primary clinical outcome measures were: (1) development of a Grade 3 or higher TEAE through 28 days; (2) detection (detectable versus undetectable) of SARS-CoV-2 RNA from NP swabs at days 3, 7, 14, 21, and 28; (3) duration of targeted COVID-19-associated symptoms from day 0 (utilizing daily diary data), where duration was defined as the number of days from day 0 to the last day on or before study day 28 when any targeted symptoms that were self-assessed as moderate or severe at day 0 (before study intervention) were still scored as moderate or severe (i.e., not mild or absent), or any targeted symptoms scored as mild or absent at day 0 were still scored as mild or worse (i.e. not absent). Participants with ongoing unimproved symptoms at day 28 were treated as having a symptom duration of 28 days for analysis.

Secondary outcome measures included all-cause hospitalization and death; quantitative NP SARS-CoV-2 RNA levels; area under the curve (AUC) of symptom scores from days 0–28; change in inflammatory markers from baseline through week 24; development of AESIs, specifically Grade 1 or higher IRRs and Grade 1 or higher allergic/hypersensitivity reactions; SAEs through day 28 and through week 24; and progression of 1 or more COVID-19-associated symptoms to a worse status than recorded in the study diary at entry. The full set of secondary and exploratory outcome measures are provided in the protocol in Supplementary Methods.

### Virology
**Sample collection and SARS-CoV-2 RNA quantitation.** NP and AN samples were collected using standardized swabs and collection procedures. AN swabs were self-collected by participants daily days 0-14. Site-collected NP swabs, AN swabs collected on site, and EDTA plasma samples were frozen and stored at −80 °C (−65 °C to −95 °C) on the day of collection. AN swabs collected remotely were stored at cool

temperatures (refrigerated or in a study-provided cooler with a combination of refrigerated and frozen gel packs) and returned to the site and frozen at −80 °C (−65 °C to −95 °C) within 7 days of collection. Samples were shipped on dry ice to a central laboratory (University of Washington) for quantitative SARS-CoV-2 RNA testing using the Abbott m2000sp/rt platform with a validated internal standard[22]. The collection, storage, processing, and assay methods have previously been validated (assay methods further described below)[22].

A modified FDA EUA-approved qualitative RealTime SARS-CoV-2 assay (Abbott Molecular, Des Plaines, IL)[23,24] was used to develop a SARS-CoV-2 quantitative laboratory developed test (LDT), using Abbott SARS-CoV-2 calibration standards that correlate cycle threshold (Ct) and viral load[22]. Identical extraction and amplification/detection protocols developed for the RealTime SARS-CoV-2 qualitative EUA assay were also used for the RealTime SARS-CoV-2 quantitative LDT. The analytical performance of the SARS-CoV-2 quantitative LDT was evaluated using commercially available SARS-CoV-2 material or using the Abbott SARS-CoV-2 material. Analytical analysis consisted of linearity, limit of detection, inter-run and intra-run reproducibility[23,24].

Open mode functionality on m2000sp/rt system instrumentation was utilized for the SARS-CoV-2 quantitative LDT. For NP and AN swabs, three-mL RPMI + 2% fetal bovine serum (FBS) was added to each tube containing a dry swab; the tube was vortexed and then incubated for 15 min at room temperature. The sample was then removed from the swab and aliquoted into a separate tube. A 500 uL aliquot of swab eluate (or plasma) was used for sample extraction; viral nucleic acid bound to the microparticles was eluted with 90 µl of elution buffer and singleton 40 µl aliquots of the eluate were used for the amplification and detection reaction using the Abbott m2000 sp/rt system[22]. The SARS-CoV-2 quantitative LDT utilized 10 unread cycles as part of the amplification and detection. In this assay, two calibrator levels (3 $\log_{10}$ RNA copies/mL and 6 $\log_{10}$ RNA copies/mL) tested in triplicate were used to generate a calibration curve and three control levels (negative, low positive at 3 $\log_{10}$ RNA copies/mL, and high positive at 5 $\log_{10}$ RNA copies/mL) and a low positive in-house control, all in singleton, were included in each run for quality management.

The assay limit of detection (LoD) was 1.4 $\log_{10}$ copies/mL, lower limit of quantification (LLoQ) was 2 $\log_{10}$ copies/mL, and upper limit of quantification (ULoQ) was 7 $\log_{10}$ copies/mL. For samples with RNA levels >ULoQ, the assay was rerun with dilutions to obtain a quantitative value.

### SARS-CoV-2 sequencing and variant analysis
S gene sequencing was performed on NP swab samples for all participants with a SARS-CoV-2 RNA level ≥2 $\log_{10}$ SARS-CoV-2 RNA copies/mL at study entry or the earliest subsequent time point. Viral RNA extraction was performed on 1 mL of swab eluate by use of the TRIzol-LS™ Reagent (ThermoFisher), as previously described[20]. cDNA synthesis was performed using Superscript IV reverse transcriptase (Invitrogen) and S gene amplification was performed using a nested PCR strategy with in-house designed primer sets targeting codons 1-814 of Spike[25]. The primer details are as follows: PCR Round 1 Fwd-primer ACAGAACATTCTTGGAATGCT, Rev-primer TCTTCAATAAATGACCTCTTGC; PCR Round 2 Fwd-primer AATCCAATTCAGTTGTCTTCC and Rev-primer TGCTTGGTTTTGATGGATCTG. Illumina library construction was performed using the Nextera XT Library Prep Kit (Illumina). Sequencing was performed on the Illumina MiSeq platform and deep sequencing data analysis was carried out using the Stanford Coronavirus Antiviral & Resistance Database (CoVDB) platform (https://covdb.stanford.edu/sierra/sars2/by-reads/?cutoff=0.01&mixrate=0.01)[26]. Input FASTQ sequence alignment with Wuhan-Hu-1 reference was done using MiniMap2 version 2.22 in CodFreq pipeline (https://github.com/hivdb/codfreq). The output of MiniMap2, an aligned SAM file, is converted to a CodFreq file by an in-house written Python script using a PySam library (version: 0.18.0) and further analyzed with the

CoVDB. SARS-COV-2 variant calling was done using 3 different variant calling platforms, namely, CoVDB[26], Scorpio call v1.2.123 (https://pangolin.cog-uk.io/)[27], and Nextclade v.1.13.2 (https://clades.nextstrain.org/)[28]. PCR and sequencing runs were performed once with the appropriate positive and negative controls.

### Serum and plasma biomarkers
Inflammatory and coagulation markers including lactate dehydrogenase (LDH) (LDHI2, cobas® analyzer, Roche Diagnostics), C-reactive protein (CRP) (Tina-quant® C-Reactive Protein IV, cobas® analyzer, Roche Diagnostics), ferritin (Elecsys® Ferritin, cobas® analyzer, Roche Diagnostics), D-dimer (HemosIL® D-dimer HS, ACL TOP 500 analyzer, Instrumentation Laboratory), prothrombin time (PT)/international normalized ratio (INR) (HemosIL® RecombiPlasTin 2G, ACL TOP 500 analyzer, Instrumentation Laboratory), activated partial thromboplastin time (aPTT) (HemosIL® SynthASil, ACL TOP 500 analyzer, Instrumentation Laboratory), and fibrinogen (HemosIL® Fibrinogen-C, ACL TOP 500 analyzer, Instrumentation Laboratory) were measured in real-time by a central clinical laboratory (PPD® Laboratory Services Global Central Labs) at days 0, 7, 14, 21, and 28 and weeks 12 and 24, per the manufacturers' protocols.

### Pharmacokinetic (PK) analysis
Blood samples for quantitation of bamlanivimab serum concentrations were collected pre-dose and at the following times after the end of infusion: 30 minutes, days 14 and 28 and weeks 12 and 24. Serum concentrations of bamlanivimab were determined using a validated hybrid LC-MS/MS method. Extraction was performed using Protein G coated magnetic beads for affinity capture of immunoglobulins, followed by sample purification, sequential reduction and alkylation of cysteines, and overnight on-beads enzymatic digestion with trypsin. A specific tryptic signature peptide from the Fab region of bamlanivimab was used for quantitation, with a stable isotope 4 labelled form used as the internal standard (IS). The signature peptide was identified and quantified using reversed-phase high-performance liquid chromatography with tandem mass spectrometry (HPLC-MS/MS) detection over an analyte concentration range of 5.00 to 500.00 µg/mL. Concentrations were calculated using peak area ratios of analyte to IS, and the linearity of the calibration curve determined using least squares regression analysis with a weighting factor of $1/x^2$. The assays were performed once on each sample. The assay has met FDA requirements for accuracy and reproducibility. Additional details of the method and its use in the first-in-human study of bamlanivimab have been published[21]. PK parameters of interest were maximum concentration (Cmax), area under the concentration-time curve from time 0 to infinity (AUC$_{0-\infty}$), elimination half-life, and total body clearance (CL), which were calculated based on the statistical moment theory using the trapezoidal rule and linear regression (WinNonLin v8.3.4, Certara, Princeton, NJ, USA).

### Power analysis and sample size calculation
The sample size of 110 participants randomized to each arm was selected to give high power to identify an active agent based on the primary virologic outcome. At the time the study was designed, there were no data in outpatients with COVID-19 to inform expected differences in proportion undetectable for NP SARS-CoV-2 RNA over 28 days. We estimated that a 20% absolute increase in the proportion undetectable would be clinically relevant, and 110 participants assigned to each arm would have 82.5 to 95.5% power dependent on the proportion undetectable in the placebo arm, with a two-sided Type I error rate of 5%.

### Statistical analysis
The analysis population included all participants who initiated study intervention (bamlanivimab or placebo). Four participants enrolled to

the 7000 mg dose cohort received 700 mg bamlanivimab or placebo and were included in the 700 mg analysis population (the randomization to active agent or placebo remained valid). One participant enrolled in the 700 mg dose cohort received 7000 mg bamlanivimab and was included in the 7000 mg analysis population.

The proportion of participants experiencing a grade 2 or higher and grade 3 or higher TEAE was compared between arms using log-binomial regression and summarized with a risk ratio (RR), corresponding 95% CI, and $p$-value based on the Wald test. The proportion of participants with undetectable SARS-CoV-2 RNA was compared between arms across study visits using Poisson regression with robust variance adjusted for baseline (day 0) $\log_{10}$ transformed SARS-CoV-2 RNA level and summarized with RR and 95% CI at each time, and Wald test across the multiple times. Quantitative SARS-CoV-2 RNA levels were compared between arms using Wilcoxon rank-sum tests, separately at each post-entry study visit, without adjustment for baseline value. For this comparison, results below the LoD were imputed as the lowest rank, and values above the LoD but below the LLoQ were imputed as the second lowest rank. For summaries of quantitative RNA levels, values below the LoD were imputed as $0.7 \log_{10}$ copies/ml (i.e., half the distance from zero to the LoD), values above the LoD but below the LLoQ were imputed as $1.7 \log_{10}$ copies/ml (i.e., half the distance from the LoD to the LLoQ), and values above the ULoQ were imputed as $8 \log_{10}$ copies/ml if a numerical value was not available.

The two dose cohorts (700 and 7000 mg) were combined for post-hoc exploratory analyses of baseline NP, AN, and plasma SARS-CoV-2 RNA levels and modeling of viral decay. Spearman correlations evaluated associations between total symptom scores and NP and AN RNA levels. Wilcoxon tests and chi-square tests were used to evaluate NP and AN RNA levels and symptom scores between subgroups. The rates of decline of NP and AN virus after study entry were quantified in separate models using Monolix 2020 (Lixoft, Antony, France). Methods for model fitting and selection are described in Supplementary Methods.

Participant-specific symptom durations and AUC of total symptom score from days 0-28 were compared between arms using a Wilcoxon rank sum test. Due to the small number of hospitalization/death events, the proportion hospitalized/dead in the bamlanivimab and placebo arms was summarized with descriptive statistics and compared between arms using Fisher's exact test as a post-hoc analysis. Change from baseline in log-transformed inflammatory and coagulation biomarker levels was compared between bamanivimab and placebo arms using Wilcoxon tests.

No adjustment was made for the multiple comparisons across outcome measures. Statistical analyses were conducted using SAS version 9.4 and R version 4.1.0. See Supplementary Methods for complete Statistical Analysis Plan.

### Reporting summary
Further information on research design is available in the Nature Research Reporting Summary linked to this article.

## Data availability
The following publicly available databases were used for viral sequence analysis in this study: Stanford Coronavirus Antiviral & Resistance Database (CoVDB) platform (https://covdb.stanford.edu/sierra/sars2/by-reads/?cutoff=0.01&mixrate=0.01), MiniMap2 version 2.22 in CodFreq pipeline (https://github.com/hivdb/codfreq), Scorpio call v1.2.123 (https://pangolin.cog-uk.io/) and Nextclade v.1.13.2 (https://clades.nextstrain.org/). The next-generation sequencing data generated in this study have been deposited on the NCBI Short Read Archive (SRA) under accession number PRJNA816433 and PRJNA859660. Other data are available under restricted access due to ethical restrictions. Access can be requested by submitting a data request at https://submit.mis.s-3.net/ and will require the written

agreement of the AIDS Clinical Trials Group (ACTG) and the manufacturer of the investigational product. Requests will be addressed as per ACTG standard operating procedures. Completion of an ACTG Data Use Agreement may be required. Aggregate data generated in this study are provided in the Source Data file. Source data are provided with this paper.

## Code availability
All analyses were performed using code available in standard software packages. No new code was developed for this manuscript. Specifics on programs used are available upon request from sdac.data@sdac.harvard.edu.

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

## Acknowledgements

We thank the study participants, site staff, site investigators, and the entire ACTIV-2/A5401 study team; the AIDS Clinical Trials Group, including Lara Hosey and Jhoanna Roa; the UW Virology Specialty Laboratory staff, including Emily Degli-Angeli, Erin Goecker, Glenda Daza, Socorro Harb, and Joan Dragavon; the ACTG Laboratory Center, including Grace Aldrovandi and William Murtaugh; Frontier Science, including Marlene Cooper and Howard Gutzman; the Harvard Center for Biostatistics in AIDS Research (CBAR) and ACTG Statistical and Data Analysis Center (SDAC); the National Institute of Allergy and Infectious Diseases (NIAID)/Division of AIDS (DAIDS), including Peter Kim; the U.S. Government Response to COVID-19, including Bill Erhardt; the Foundation for the National Institutes of Health and the Accelerating COVID-19 Therapeutic Interventions and Vaccines (ACTIV) partnership, including Stacey Adams; and the PPD clinical research business of Thermo Fisher Scientific. We also thank the members of the ACTIV-2/A5401 data and safety monitoring board—Graeme A. Meintjes, PhD, MBChB (Chair), Barbara E. Murray, MD, Stuart Campbell Ray, MD, Valeria Cavalcanti Rolla, MD, PhD, Haroon Saloojee, MBBCh, FCPaed, MSc, Anastasios A. Tsiatis, PhD, Paul A. Volberding, MD, Jonathan Kimmelman, PhD, David Glidden, PhD, and Sally Hunsberger, PhD (Executive Secretary). This work was supported by the National Institute of Allergy and Infectious Diseases of the National Institutes of Health under Award Number UM1AI068634 (MDH), UM1AI068636 (JSC), and UM1AI106701 (GA). The content is solely the responsibility of the authors and does not necessarily represent the official views of the National Institutes of Health. Portions of this work were performed at Los Alamos National Laboratory under the auspices of the US Dept. of Energy under contract 89233218CNA000001 and supported by NIH grants R01-AI028433 (ASP), R01-OD011095 (ASP), and Los Alamos National Laboratory LDRD 20200743ER, 20200695ER, and 20210730ER. Study medication was donated by Eli Lilly and Company. The study sponsor, the NIH Division of AIDS, participated in the design of the study and reviewed and approved the protocol prior to study initiation. Oversight and responsibility for data collection and primary data analyses were delegated by the sponsor to PPD clinical research, a Contract Research Organization (CRO). Safety laboratories and inflammatory and coagulation biomarkers were measured at PPD Laboratory Services Global Central Labs and statistical analyses were performed by the CRO. A sponsor representative (ACJ) reviewed and approved the manuscript. Lilly voluntarily asked the FDA to revoke the Emergency Use Authorization (EUA) for bamlanivimab 700 mg alone in April 2021. This request was not due to any new safety concerns.

## Author contributions

K.W.C., C.M., E.S.D., D.A.W., J.Z.L., A.C.J., P.K., K.P., A.N., W.F., C.V.F., J.J.E., J.S.C., M.D.H., and D.M.S. conceived and designed the research. K.W.C., E.S.D., D.A.W., C.M., M.C.C., R.D., and R.C. generated data. J.R., M.G., Y.L., V.B., R.M.R., A.S.P., C.V.F., and M.D.H. analyzed the data. K.W.C., E.S.D., D.A.W., J.Z.L., R.C., W.F., R.M.R., A.S.P., C.V.F., J.J.E., J.S.C., M.D.H., and D.M.S. interpreted and wrote the paper.

## Competing interests

K.W.C. has received research funding to the institution from Merck Sharp & Dohme and is a consultant for Pardes Biosciences. E.S.D. receives consulting fees from Gilead Sciences, Merck, and GSK/ViiV and research support through the institution from Gilead Sciences and GSK/ViiV. D.A.W. has received funding to the institution to support research and honoraria for advisory boards and consulting from Gilead Sciences. J.Z.L. has consulted for Abbvie. C.M. has received research funding to the institution from E.L. P.K., K.P., and A.N. are employees and shareholders of E.L. W.F. has received research funding to the institution from Ridgeback Biopharmaceuticals, served on adjudication committees for Janssen, Syneos, and consulted for Roche and Merck. J.J.E. is an ad hoc consultant to GSK/VIR, data monitoring committee (DMC) chair for Adagio Phase III studies. J.S.C. has consulted for Merck and Company. D.M.S. has consulted for the following companies Fluxergy, Kiadis, Linear Therapies, Matrix BioMed, Arena Pharmaceuticals, VxBiosciences, Model Medicines, Bayer Pharmaceuticals, Signant Health and Brio Clinical. All other authors (R.C., C.M., J.R., M.G., A.C.J., Y.L., M.C.C., R.D., V.B., R.M.R., A.S.P., C.V.F., and M.D.H.) report no competing interests.
