## [Peer Review File · Nature Communications]

REVIEWERS' COMMENTS

Reviewer #3 (Remarks to the Author):

The manuscript reports on a phase 2 clinical trial aimed at evaluating the efficacy of i.v. administered monoclonal antibody against SARS-CoV-2 (Bamlanivimab) in outpatients upon infection. No changes in symptoms duration despite lower virus RNA levels were observed. The clinical study appears well performed but provides very limited biological insights. The interest and potential impact of this report is diminished by the fact that this monoclonal antibody is not efficacious against the omicron variant, and that the FDA has withdrawn its emergency use authorization.

Reviewer #4 (Remarks to the Author):

The authors have addressed my previous concerns and I have no further comments.

Response to Reviewers' Comments

Thank you for the opportunity to further revise our manuscript. A point-by-point response to the reviewers' comments is provided below.

Reviewer #3 (Remarks to the Author):

The manuscript reports on a phase 2 clinical trial aimed at evaluating the efficacy of i.v. administered monoclonal antibody against SARS-CoV-2 (Bamlanivimab) in outpatients upon infection. No changes in symptoms duration despite lower virus RNA levels were observed. The clinical study appears well performed but provides very limited biological insights. The interest and potential impact of this report is diminished by the fact that this monoclonal antibody is not efficacious against the omicron variant, and that the FDA has withdrawn its emergency use authorization.

Response: While bamlanivimab is not currently used in clinical practice, we believe this study remains clinically relevant and provides insights not available from other studies of COVID-19 therapeutics to date. The data are particularly rich and robust as the study was carefully designed, including the randomization scheme, to consider the impact of host characteristics and disease stage on virologic and clinical outcomes, with intensive blood and respiratory tract sampling built into the rigorous framework of a randomized controlled trial. The discussion has been revised to more fully describe limitations of the study, while also highlighting the findings that are relevant to understanding SARS-CoV-2 disease pathogenesis and the design of future COVID-19 therapeutics (mAb and small molecule) and studies. Furthermore, the use of mAb regimens is directed by the current circulating variants, and it is possible that bamlanivimab may return to clinical use in the future, as has already occurred once, when Delta became the dominant variant. Finally, the identical molecular targets and mechanisms of action of all SARS-CoV-2-targeting monoclonal antibodies make it likely that these findings will translate to the other mAbs that are in use today, including single mAb regimens for COVID-19 therapeutics.

Reviewer #4 (Remarks to the Author):

The authors have addressed my previous concerns and I have no further comments.

Response: We thank the reviewer for their previous comments.